# Radar Maneuvering Target Detection Based on Product Scale Zoom Discrete Chirp Fourier Transform

**Lang Xia** [1], **Huotao Gao** [1,*], **Lizheng Liang** [2], **Taoming Lu** [1] **and Boning Feng** [1]

1    Electronic Information School, Wuhan University, Wuhan 430072, China
2    Mingfei Weiye Technology Co., Ltd., Wuhan 430061, China
*    Correspondence: ght@whu.edu.cn

**Abstract:** Long-time coherent integration works to significantly increase the detection probability for maneuvering targets. However, during the observation time, the problems that often tend to occur are range cell migration (RCM) and Doppler frequency cell migration (DFCM), due to the high velocity and acceleration of the maneuvering target, which can reduce the detection of the maneuvering targets. In this regard, we propose a new coherent integration approach, based on the product scale zoom discrete chirp Fourier transform (PSZDCFT). Specifically, by introducing the zoom operation into the modified discrete chirp Fourier transform (MDCFT), the zoom discrete chirp Fourier transform (ZDCFT) can correctly estimate the centroid frequency and chirp rate of the linear frequency-modulated signal (LFM), regardless of whether the parameters of the LFM signal are outside the estimation scopes. Then, the scale operation, combined with ZDCFT, is performed on the radar echo signal in the range frequency slow time domain, to remove the coupling. Thereafter, a product operation is executed along the range frequency to inhibit spurious peaks and reinforce the true peak. Finally, the velocity and acceleration of the target estimated from the true peak position, are used to construct a phase compensation function to eliminate the RCM and DFCM, thus achieving coherent integration. The method is a linear transform without energy loss, and is suitable for low signal-to-noise (SNR) environments. Moreover, the method can be effectively fulfilled based on the chirp-z transform (CZT), which prevents the brute-force search. Thus, the method reaches a favorable tradeoff between anti-noise performance and computational load. Intensive simulations demonstrate the effectiveness of the proposed method.

**Keywords:** maneuvering target; coherent integration; product scale zoom discrete chirp Fourier transform; range cell migration; doppler frequency cell migration

## 1. Introduction

In recent years, radar maneuvering target detection has attracted extensive research interest in spaceborne, airborne, and ground-based radar [1–3], due to the growing demand for practical applications, such as for high-resolution remote sensing imaging and high precision tracking [4–8]. However, it is often difficult for radar to detect high maneuvering targets, due to the weak radar echoes caused by their low radar cross section (RCS) [9–13]. Long-time coherent integration can enhance the signal-to-noise ratio (SNR) of the echo signal and increase the detection probability of maneuvering targets [14–16]. However, during coherent processing, the high velocity and acceleration of the maneuvering target can induce linear range cell migration (LRCM), quadratic range cell migration (QRCM), and Doppler frequency cell migration (DFCM). Not only that, when the radar pulse repetition frequency (PRF) is low, the excess motion parameters of the maneuvering target may not be estimated correctly. These adverse effects can seriously degrade the detection performance of traditional methods (e.g., moving target detection, MTD). Therefore, in order to improve the detection performance, it is imperative to propose an effective method to solve these problems.

For the elimination of LRCM, many successful methods have been proposed. Representative methods involve the keystone transform (KT) [17,18], Radon Fourier transform (RFT) [19,20], axis rotation transform (ART) [21], scaled inverse Fourier transform (SIFT) [22], and frequency-domain deramp-keystone transform (FDDKT) [23], where the KT is a linear transform for eliminating LRCM of multiple targets without prior information. However, the KT may suffer from Doppler ambiguity, due to the high velocity and low PRF. The RFT and ART eliminate the LRCM through the two-dimensional search procedure, which may require a heavy computational burden. The SIFT and FDDKT can achieve the cancellation of LRCM with the help of the symmetric autocorrelation function, which can maintain a low computational burden at the cost of anti-noise performance. Nevertheless, the coherent gain of the above method can decline sharply, due to overlooking the QRCM and DFCM arising from the target's acceleration.

To address this issue, various methods have been introduced. The generalized RFT (GRFT) [24], is a classical method which extracts the target trajectory through a three-dimensional search, and constructs a corresponding filter to accomplish coherent integration. However, the consequent problem is that the GRFT has severe blind speed side lobes (BSSLs) interference, due to the nature of its filter construction. Based on this, the Radon-Lv's distribution (RLVD) [25] and Radon-fractional Fourier transform (RFRFT) [26] were developed, and they can successfully avert the effect of BSSLs due to their adoption of LVD and FRFT as filters, respectively. Unfortunately, the aforementioned methods are prohibitively computationally intensive and are not conducive to real-time processing. In this respect, improved axis rotation fractional Fourier transform (IAR-FRFT) [27], modified axis rotation transform and Lv's transform (MART-LVT) [28], and modified location rotation transform and improved discrete chirp Fourier transform (MLRT-IDCFT) [29], can slightly diminish the computational burden, by reducing the search dimension. However, these three methods come at the cost of not being able to eliminate the QRCM, which may degrade some of the accumulation gain. Moreover, the computational burden of these methods remain considerable. To further drastically reduce the computational cost for real-time processing, some rank-reduction based methods have been widely investigated. Typical methods include, the adjacent cross-correlation function (ACCF) [30], three-dimensional scaled transform (TDST) [31], second-order Wigner–Ville distribution (SoWVD) [32], and frequency autocorrelation function and Lv's distribution (FAF-LVD) [33]. These methods can reduce the rank of the echo signal by carrying out correlation operations, thus avoiding the brute-force search and reducing the computational load. Although these methods have low computational complexity and facilitate real-time processing, they are only suitable for high SNR environments due to the loss of signal energy and cross terms caused by correlation operations. Besides, time-reversal transform (TRT)-based methods, such as improved axis rotation and time reversal transform (IAR-TRT) [34], keystone transform and time reversal transform (KT-TRT) [35], and phase compensation and time-reversal transform (PC-TRT) [36], can also obtain efficient calculations, by implementing TRT operations. However, similar to the rank-reduction-based approaches, the anti-noise performance of TRT-based methods can also drop tremendously in a low SNR environment, due to the fact that TRT operations are nonlinear.

With the aim of striking a better balance between computational cost and anti-noise performance, this paper proposes a novel coherent integration approach, based on the product scale zoom discrete chirp Fourier transform (PSZDCFT). The basic idea of the method, is that the velocity and acceleration of the maneuvering target are estimated via PSZDCFT, then the phase compensation function is constructed to eliminate the RCM and DFCM, and finally, the slow-time Fourier transform is applied, to accomplish coherent integration. The zoom operation is used to extend the estimation scopes of the modified discrete chirp Fourier transform (MDCFT) [37] when the velocity and acceleration of the target are superabundant, which also results in the occurrence of other false peaks in the two-dimensional spectrum. The scale operation is used to eliminate the coupling between the true peak location and the range frequency, while making the false peak locations

couple with the range frequency. The product operation is used to identify and reinforce the true peak, while suppressing the false peaks. The most important features of the proposed method are, extending the estimation ranges of MDCFT by zoom operation, and improving the parameter estimation performance by product operation. The former can be applied to the uniform motion target with velocity ambiguity, and the latter can be developed into other parameter estimation methods, such as dechirp estimation. Moreover, in contrast to the rank-reduction-based and time-reverse-based methods, the proposed method is a linear transform without loss of target energy and no cross terms for multiple targets. Compared with the traditional methods of brute-force search, such as GRFT and RLVD, this method is search-free, thanks to its efficient implementation by fast Fourier transform (FFT), inverse fast Fourier transform (IFFT), and complex multiplication, which greatly reduces the computational burden, without significant degradation in detection performance. Thus overall, the presented approach reaches a favorable tradeoff between anti-noise performance and computational complexity.

The rest of this paper is arranged as follows. In Section 2, the signal model is formulated and the problem to be addressed is presented. Section 3 details the development and rationale of the proposed method. In Section 4, the relevant analysis of the proposed approach is given. Section 5 provides the simulation results. Finally, the conclusions are drawn in Section 6.

## 2. Signal Model and Problem Formulation

Assume that a pulsed radar transmits a narrow-band linear frequency-modulated signal (LFM), as follows

$$s(\hat{t}, \eta_m) = \text{rect}\left(\frac{\hat{t}}{T_p}\right) \exp\left(j\pi\gamma\hat{t}^2\right) \exp\left[j2\pi f_c(\hat{t} + \eta_m)\right], \tag{1}$$

where $T_p$ and $\text{rect}(\cdot)$ represent the pulse width and rectangular function, respectively. $\gamma$ and $\hat{t}$ are the frequency modulated rate and fast time, respectively. $f_c$ is the carrier frequency. $\eta_m = m/f_r$ ($m = -M/2, \ldots, -1, 0, 1, \ldots, M/2 - 1$) is the slow time. $f_r = 1/T_r$ and $M$ indicate the PRF and pulse number, respectively. $T_r$ is the pulse repetition interval (PRI).

We consider a second-order motion model and assume that the instantaneous slant range $R(\eta_m)$, between the target and the radar is

$$R(\eta_m) = R_0 + v_0\eta_m + \frac{1}{2}a_0\eta_m^2, \tag{2}$$

where $R_0$, $v_0$, and $a_0$ are the initial range, radial velocity, and acceleration of the target, respectively.

Then, after the signal demodulation, the received baseband signal is stated as

$$\begin{aligned} s_r(\hat{t}, \eta_m) = \ &\sigma_0 \, \text{rect}\left[\frac{\hat{t} - 2R(\eta_m)/c}{T_p}\right] \\ &\times \exp\left[-j\frac{4\pi f_c}{c}R(\eta_m)\right] \\ &\times \exp\left\{j\pi\gamma\left[\hat{t} - \frac{2R(\eta_m)}{c}\right]^2\right\}, \end{aligned} \tag{3}$$

where $\sigma_0$ and $c$ indicate the echo amplitude and velocity of light, respectively. After pulse compression (PC) using matched filters, the received signal can be noted as

$$s_c(\hat{t}, \eta_m) = \sigma_1 \, \text{sinc}\left[B\left(\hat{t} - \frac{2R(\eta_m)}{c}\right)\right] \exp\left(-j\frac{4\pi f_c R(\eta_m)}{c}\right), \tag{4}$$

where $\sigma_1$ and $B$ are the signal amplitude after PC and the signal bandwidth, respectively.

Then, performing FFT on Equation (4) along the fast time $\hat{t}$, yields

$$S_c(f, \eta_m) = \sigma_2 \, \text{rect}\left(\frac{f}{B}\right) \exp\left[-j\frac{4\pi(f+f_c)R(\eta_m)}{c}\right], \tag{5}$$

where $f$ is the range frequency corresponding to the fast time $\hat{t}$, and $\sigma_2$ indicates the signal amplitude after performing FFT.

Substituting Equation (2) into Equation (5), one obtains

$$S_c(f, \eta_m) = \sigma_3 \exp\left[-j4\pi\frac{f}{c}\left(R_0 + v_0\eta_m + \frac{1}{2}a_0\eta_m^2\right)\right]$$
$$\times \exp\left[-j\frac{4\pi}{\lambda}\left(R_0 + v_0\eta_m + \frac{1}{2}a_0\eta_m^2\right)\right], \tag{6}$$

where $\sigma_3 = \sigma_2 \, \text{rect}\left(\frac{f}{B}\right)$. $\lambda = c/f_c$ is the wavelength.

Since the range frequency is coupled to slow time, the first exponential term of Equation (6) could bring about the LRCM due to velocity and QRCM due to acceleration. The second exponential term may result in the DFCM induced by acceleration. Therefore, it is critical to accurately estimate the velocity and acceleration of the maneuvering target, to eliminate the adverse effects of the RCM and DFCM and improve the detection performance.

## 3. Description of the Proposed Method

### 3.1. MDCFT and Its Limitations

In [38], Xia proposes the DCFT to estimate the centroid frequency and chirp rate of the LFM signal. However, this method has the requirement that the total sampling number of the signal is prime and the parameters of the signal are integers. With this in mind, the MDCFT is proposed, which eliminates two constraints of DCFT. The principle of MDCFT is as follows.

The analog form of the LFM signal can be expressed as

$$z(\eta_m) = \sigma \exp\left[j2\pi\left(f_0\eta_m + \mu_0\eta_m^2\right)\right], \tag{7}$$

where $\sigma$ and $\eta_m$ denote the signal amplitude and slow time, respectively. $f_0$ and $u_0$ are the centroid frequency and chirp rate, respectively. After discretization, Equation (7) can be written as

$$z(m) = \sigma \exp\left[j2\pi\left(f_0 T_r m + \mu_0 T_r^2 m^2\right)\right]$$
$$= \sigma W_M^{-\left(k_0 m + \frac{l_0}{M}m^2\right)}, \tag{8}$$

where $k_0 = f_0 T_r M$ and $l_0 = \mu_0 T_r^2 M^2$ represent the digital centroid frequency and chirp rate of the signal, respectively. $W_M = exp(-j2\pi/M)$.

The MDCFT of $z(m)$ is defined as

$$Z_{MDCFT}(k, l) = \frac{1}{\sqrt{M}} \sum_{m=-M/2}^{M/2-1} z(m) W_M^{\left(km + \frac{l}{M}m^2\right)}, \tag{9}$$

where $k \in [-M/2, M/2-1]$ and $l \in [-M/2, M/2-1]$ denote the centroid frequency index and chirp frequency index, respectively. It is obvious that the estimation scopes of both $k_0$ and $l_0$ are $[-M/2, M/2)$. With the substitution of Equation (8) into Equation (9), we can obtain

$$Z_{MDCFT}(k, l) = \sigma \frac{1}{\sqrt{M}} \sum_{m=-M/2}^{M/2-1} W_M^{\frac{1}{M}(l-l_0)m^2} W_M^{(k-k_0)m}. \tag{10}$$

Depending on whether $k_0$ and $l_0$ exceed the estimation scopes, the peak position of $|Z_{MDCFT}(k, l)|$ can be divided into the following cases.

Case 1: $|k_0| < M/2$ and $|l_0| < M/2$. This means that both $k_0$ and $l_0$ are within their estimation scopes, and as shown in Equation (10), the peak of the two-dimensional spectrum can be observed at $(k_0, l_0)$.

Case 2: $|k_0| \geq M/2$ and $|l_0| < M/2$. In this case, $k_0$ is over its valid estimation range, which implies that Doppler ambiguity would occur. $k_0$ can be written as

$$k_0 = k_0' + pM, \tag{11}$$

where $p = round(k_0/M)$ is the number of the Doppler ambiguity. $k_0' \in (-M/2, M/2)$. Inserting Equation (11) into Equation (10), one obtains

$$Z_{MDCFT}(k, l) = \sigma \frac{1}{\sqrt{M}} \sum_{m=-M/2}^{M/2-1} W_M^{\frac{1}{M}(l-l_0)m^2} W_M^{(k-k_0')m}. \tag{12}$$

Here, we use the constant equation $W_M^{-pMm} = 1$. As shown in Equation (12), we can see that the peak is located at $(k_0', l_0)$ in the two-dimensional spectrum, which is the folded peak. From the coordinates of the folded peak, we cannot get the correct digital centroid frequency.

Case 3: $M/2 \leq |l_0| < M^2/2$. At this point, since $l_0$ is out of the estimation scope, $k_0$ and $l_0$ cannot be accurately estimated, regardless of the value of $k_0$. In the two-dimensional spectrum, $|Z_{MDCFT}(k, l)|$ has no distinctive sharp peak.

Case 4: $M^2/2 \leq |l_0|$. In this case, $l_0$ is still out of the estimation scope, but similar to case 2, chirp rate ambiguity can occur. $l_0$ can be written as

$$l_0 = l_0' + hM^2, \tag{13}$$

where $h = round(l_0/M^2)$ is the number of the chirp rate ambiguity . $l_0' \in (-M^2/2, M^2/2)$. By taking Equation (13) into Equation (10), we have

$$Z_{MDCFT}(k, l) = \sigma \frac{1}{\sqrt{M}} \sum_{m=-M/2}^{M/2-1} W_M^{\frac{1}{M}(l-l_0')m^2} W_M^{(k-k_0)m}. \tag{14}$$

$W_M^{-\frac{1}{M}hM^2m^2} = 1$ is used here. The analysis of Equation (14) is the same as the previous three cases, and it is worth noting that the correct value of $l_0$ cannot be obtained in this case. In practice, chirp rate ambiguity is almost impossible, that is, $|l_0| \ngeq M^2/2$. Therefore, we will not analyze this case later.

### 3.2. Zoom DCFT

In the previous subsection, we expounded the principle and limitations of the MDCFT. In order to remove these limitations and make MDCFT more practical, we proposed the zoom DCFT method (ZDCFT).

The ZDCFT of $z(m)$ is defined as follows

$$Z_{ZDCFT}(k, l) = \text{ZDCFT}[z(m)]$$
$$= \frac{1}{\sqrt{M}} \sum_{m=-M/2}^{M/2-1} z(m) W_M^{\left(\alpha km + \beta \frac{l}{M}m^2\right)}, \tag{15}$$

where $k \in [-M/2, M/2)$ and $l \in [-M/2, M/2)$ represent the centroid frequency index and chirp frequency index, respectively. $\alpha \in \mathbf{N}^*$ and $\beta \in \mathbf{N}^*$ denote the zoom factors of the centroid frequency and chirp rate, respectively.

In order to clarify the role of the ZDCFT, we analyze the first three cases in the previous subsection one by one.

Inserting Equation (8) into Equation (15), we obtain

$$Z_{ZDCFT}(k,l) = \sigma \frac{1}{\sqrt{M}} \sum_{m=-M/2}^{M/2-1} W_{M_l}^{(l-l_0/\beta)m^2} W_{M_k}^{(k-k_0/\alpha)m}, \tag{16}$$

where $M_l = M^2/\beta$ and $M_k = M/\alpha$ are the new implied periods of $Z_{ZDCFT}(k,l)$ in the $k$ and $l$ domains, respectively.

For case 1, i.e., $|k_0| < M/2$ and $|l_0| < M/2$, we should just take $\alpha = \beta = 1$. Then, the ZDCFT degenerates into the MDCFT. The peak of the theoretical two-dimensional spectrum is located at $(k_0, l_0)$.

For case 2, i.e., $|k_0| \geq M/2$ and $|l_0| < M/2$, we know that $|l_0| < M/2$, so $\beta$ is taken as 1. Since $|k_0| \geq M/2$ exceeds the estimation scope, we should take the appropriate $\alpha$ ($\alpha > 1$) to extend the estimation scope of $k_0$, i.e.,

$$[-\frac{M}{2}\alpha, \frac{M}{2}\alpha). \tag{17}$$

In other words, it is to make $k_0/\alpha$ lie within the computational range of $k$, i.e., $k_0/\alpha \in [-M/2, M/2]$. However, the ensuing problem is that $|Z_{MDCFT}(k,l)|$ would generate $\alpha$ peaks in the $k$ domain. This is because the length of the computational interval in the k-domain is $\alpha$ times longer than its implied period by the zoom operation, i.e., $M/M_k = \alpha$. Therefore, with $\beta = 1$, Equation (16) can be rewritten as follows

$$Z_{ZDCFT}(k,l) = \sigma \frac{1}{\sqrt{M}} \sum_{m=-M/2}^{M/2-1} W_{M_l}^{(l-l_0)m^2} W_{M_k}^{(k-k_0/\alpha-FM_k)m}, \tag{18}$$

where $F \in \mathbf{Z}$, $M_l = M^2$, and $M_k = M/\alpha$. It can be seen from Equation (18) that these $\alpha$ peaks are located at

$$\begin{cases} \hat{k} = k_0/\alpha + FM_k \\ \hat{l} = l_0 \\ \hat{k} \in [-M/2, M/2) \\ \hat{l} \in [-M/2, M/2) \\ F \in \mathbf{Z} \end{cases} \tag{19}$$

When $F = 0$, we can obtain the correct digital centroid frequency and chirp rate, i.e., $(\hat{k}_0, \hat{l}_0) = (\alpha\hat{k}, \hat{l})$.

**Example 1.** *Figure 1 illustrates the role of the zoom factor of the centroid frequency $\alpha$. The simulation parameters of the LFM signal are: $\mu_0 = 45$ Hz/s, $f_0 = 630$ Hz, $f_r = 400$ Hz, and $M = 800$. Then, we can get $k_0 = f_0 M/f_r = 1260$ and $l_0 = \mu_0 M^2/f_r^2 = 180$. It can be seen that $k_0$ is out of the estimation scope while $l_0$ is not. Therefore, we choose $\alpha = 4$ and $\beta = 1$, which means that four peaks would be observed in the two-dimensional spectrum, of which the true peak is located at (315, 180). With the help of the new implied period $M_k = M/\alpha = 200$, we can also derive the theoretical x-coordinate of one of the false peaks according to Equation (19), i.e., $-85 = 315 - 2 \times 200$. The value of the y-coordinate is the same as the true peak. The result of MDCFT is given in Figure 1a. It can be seen that, although there is a peak in the two-dimensional spectrum, the x-coordinate of this peak is the folded digital centroid frequency, i.e., $k_0' = -340$. Figure 1b shows the result of ZDCFT, which is in agreement with the theory.*

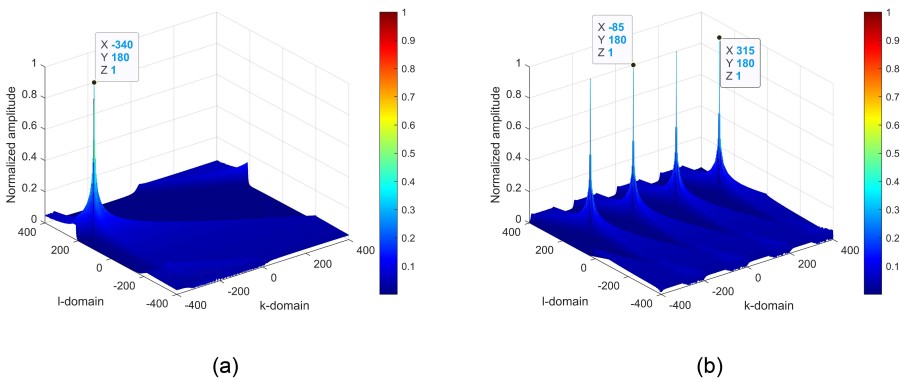

**Figure 1.** Simulation results of Example 1. (**a**) The result of MDCFT. (**b**) The result of ZDCFT ($\alpha = 4, \beta = 1$).

For case 3, i.e., $M/2 \leq |l_0| < M^2/2$, as in case 2, we should select a suitable $\beta$ ($\beta > 1$) to make $|l_0/\beta|$ lie within $M/2$. To put it another way, the estimation scope of $l_0$ is extended by $\beta$, i.e.,

$$[-\frac{M}{2}\beta, \frac{M}{2}\beta). \tag{20}$$

Note that, it is sufficient that $\beta$ is less than $M$ according to Equation (20) and that $|l_0|$ is less than $M^2/2$. Thus, unlike the analysis in case 2, the length of the computational interval in the $l$ domain is still less than its new implied period, i.e., $M < M_l = M^2/\beta$, s.t. $\beta < M$. From another perspective, the computational range of $l$ is a subset of $[-M_l/2, M_l/2)$, i.e., $[-M/2, M/2) \subset [-M_l/2, M_l/2)$, which implies that $|Z_{MDCFT}(k, l)|$ would generate only one peak in the $l$ domain, independent of $\beta$. Considering $k_0$, Equation (16) is rewritten as follows

$$Z_{ZDCFT}(k, l) = \sigma \frac{1}{\sqrt{M}} \sum_{m=-M/2}^{M/2-1} W_{M_l}^{(l-l_0/\beta)m^2} W_{M_k}^{(k-k_0/\alpha-FM_k)m}, \tag{21}$$

where $M_l = M^2/\beta$, and $M_k = M/\alpha$. From Equation (21), we can see that in the two-dimensional spectrum, there are still $\alpha$ peaks, which appear at

$$\begin{cases} \hat{k} = k_0/\alpha + FM_k \\ \hat{l} = l_0/\beta \\ \hat{k} \in [-M/2, M/2) \\ \hat{l} \in [-M/2, M/2) \\ F \in \mathbf{Z} \end{cases} \tag{22}$$

When $F = 0$, we can obtain the correct digital centroid frequency and chirp rate, i.e., $(\hat{k}_0, \hat{l}_0) = (\alpha\hat{k}, \beta\hat{l})$.

**Example 2.** *In Figure 2, we show the joint action of the zoom factors $\alpha$ and $\beta$. The simulation parameters for the LFM signal are set as: $\mu_0 = 145$ Hz/s, $f_0 = 630$ Hz, $f_r = 400$ Hz, and $M = 800$. Then, both $k_0$ and $l_0$ are out of the estimation scopes, i.e., $k_0 = 1260 > M/2 = 400$ and $l_0 = 580 > M/2$. At this point, we have that $\alpha$ equals 4 and $\beta$ equals 2, which indicates that there would be four peaks along the k-dimension and one peak along the l-dimension, in the two-dimensional spectrum. The true peak should be formed at (315, 290) in the two-dimensional spectrum. Similar to Example 1, the location of one of the three false peaks should appear at ($-85$, 290), based on Equation (22). Figure 2a shows the correct result of ZDCFT, in accordance with the theoretical analysis. For comparison, Figure 2b,c display the wrong results of ZDCFT, when $\alpha = 1$ and $\beta = 2$ and when $\alpha = 4$ and $\beta = 1$, respectively. It is evident that the former ($\alpha = 1, \beta = 2$) is blurred in the Doppler domain, while the latter ($\alpha = 4, \beta = 1$) is invalid because the chirp rate cannot be correctly estimated.*

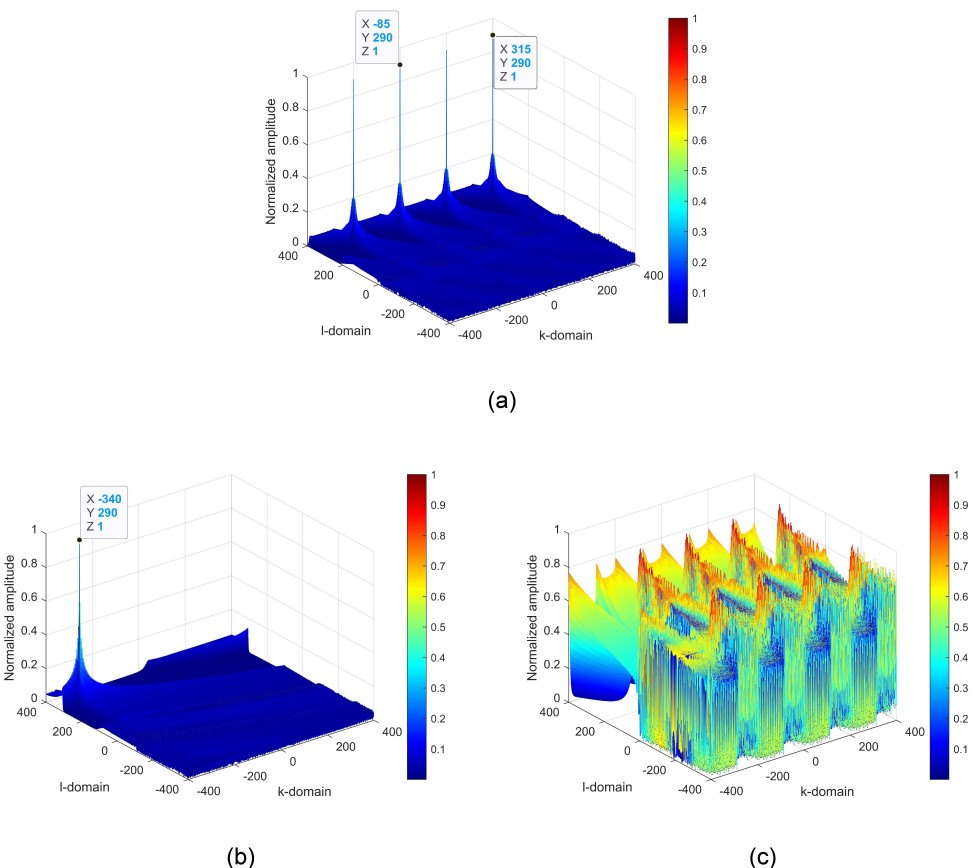

**Figure 2.** Simulation results of Example 2. (**a**) The correct result of ZDCFT ($\alpha = 4, \beta = 2$). (**b**) The blurred result of ZDCFT ($\alpha = 1, \beta = 2$). (**c**) The invalid result of ZDCFT ($\alpha = 4, \beta = 1$).

**Remark 1.** *The zoom factors for $\alpha$ and $\beta$ should be as small as possible, with the guarantee that the estimation ranges contain $k_0$ and $l_0$ [39]. The reason for this is that the spatial resolutions in the $k$ and $l$ domains are proportional to $\alpha$ and $\beta$, respectively, i.e., $\rho_k = \alpha/M$ and $\rho_l = \beta/M^2$.*

### 3.3. Scale ZDCFT

The previous subsection shows that ZDCFT can obtain correct parameter estimations, regardless of whether the centroid frequency and chirp rate of the LFM signal exceed the estimation scopes of MDCFT. For the two-dimensional radar echo signal in the range frequency-slow time domain, it is an LFM signal in the slow-time dimension with a certain range frequency cell. Therefore, we can use the ZDCFT to estimate the velocity and acceleration of the maneuvering target. However, we may obtain different values of the velocity and acceleration for different range frequency cells via ZDCFT. Considering this problem, we propose the scale ZDCFT method (SZDCFT).

The discretized form of Equation (6), for the slow-time variable $\eta_m$, is as follows

$$
\begin{aligned}
S_c(f, m) &= \sigma_3 \exp\left[-j2\pi\psi(f)\left(\frac{2R_0}{\lambda} + \frac{2v_0}{\lambda}T_r m + \frac{a_0}{\lambda}T_r^2 m^2\right)\right] \\
&= \sigma_4 W_M^{-\psi(f)\left(K_0 m + \frac{L_0}{M}m^2\right)},
\end{aligned}
\tag{23}
$$

where $\sigma_4 = \sigma_3 \exp\left(-j\frac{4\pi}{\lambda}\psi(f)R_0\right)$ denotes the complex amplitude of the signal. $K_0 = -\frac{2v_0}{\lambda}MT_r$ and $L_0 = -\frac{a_0}{\lambda}M^2T_r^2$ represent the digital centroid frequency and chirp rate of the target, respectively. $\psi(f) = 1 + f/f_c$ is the scale factor.

Then, the SZDCFT of Equation (23) is defined as

$$
\begin{aligned}
S_{SZDCFT}(f,k,l) &= \text{SZDCFT}[S_c(f,m)] \\
&= \frac{1}{\sqrt{M}} \sum_{m=-M/2}^{M/2-1} S_c(f,m) W_M^{\psi(f)\left(\alpha km + \beta \frac{l}{M} m^2\right)} \\
&= \sigma_4 \frac{1}{\sqrt{M}} \sum_{m=-M/2}^{M/2-1} W_{M_{lf}}^{(l-L_0/\beta)m^2} W_{M_{kf}}^{(k-K_0/\alpha - FM_{kf})m},
\end{aligned}
\tag{24}
$$

where $M_{lf} = \frac{M^2}{\beta\psi(f)}$ and $M_{kf} = \frac{M}{\alpha\psi(f)}$ denote the implied periods of $S_{SZDCFT}(f,k,l)$ in the $l$ and $k$ domains associated with the range frequency, respectively.

Similarly, in the two-dimensional spectrum, we can observe from Equation (24) that there are $\alpha$ peaks, which are situated at

$$
\begin{cases}
\hat{k} = K_0/\alpha + FM_{kf} \\
\hat{l} = L_0/\beta \\
\hat{k} \in [-M/2, M/2) \\
\hat{l} \in [-M/2, M/2) \\
F \in \mathbf{Z}
\end{cases}
\tag{25}
$$

In Equation (25), the true peaks will appear at the location of $(K_0/\alpha, L_0/\beta)$ when $F = 0$, which are irrespective of the range frequency. Therefore, it can be seen that the SZDCFT can neutralize the effect of the range frequency.

**Example 3.** *Figure 3 simulates the results of SZDCFT for two different scale factors, $\psi_1 = 1.01$ and $\psi_2 = 0.99$. The parameters of the LFM signal are the same as in Example 2. Correspondingly, the zoom factors of the centroid frequency $\alpha$, and chirp rate $\beta$, are equal to 4 and 2, respectively. In the two-dimensional spectrum, the true peak would be formed at (315, 290). The results of SZDCFT for the scale factors $\psi_1$ and $\psi_2$, are depicted in Figure 3a,b, respectively, which are consistent with the theoretical derivation. As a comparison, Figure 3c,d show the parameter estimations via ZDCFT, in which it can be seen that the coordinates of the peak are coupled to the scale factors.*

**Remark 2.** *It should be noted that the selection criteria for the values of $\alpha$ and $\beta$, are the same as that of the ZDCFT. This is because, although the existence of the scale factor $\psi(f)$ could make the implied periods of the SZDCFT different from those of the ZDCFT, fortunately, for the narrowband radar system, there are $f \ll f_c$ and $\psi(f) \approx 1$. Therefore, when judging whether the parameters to be estimated exceed the estimation scopes of SZDCFT, the changes in the implied periods caused by the scale factor $\psi(f)$ can be ignored.*

Here, we give more specific selection criteria for the values of $\alpha$ and $\beta$, combined with the target motion parameters.

The velocity of the maneuvering target can be written as

$$
v_0 = N_{amb}v_{amb} + v_{res},
\tag{26}
$$

where $N_{amb}$ and $v_{amb} = \frac{\lambda f_r}{2}$ are the ambiguity integers and blind speed, respectively. $|v_{res}| < v_{amb}/2$ is the unambiguous velocity.

When the target's velocity is not ambiguous, i.e., $N_{amb} = 0$ and $|v_0| = |v_{res}| < v_{amb}/2$, then we have $|K_0| = \left|\frac{2v_{res}}{\lambda}MT_r\right| < M/2$. Evidently, $\alpha$ is taken as 1.

When the target's velocity is ambiguous, i.e., $N_{amb} \neq 0$, we have $K_0 = -\frac{2v_0}{\lambda}MT_r = -N_{amb}M - K_0'$, where $K_0' = \frac{2v_{res}}{\lambda}MT_r \in (-M/2, M/2)$. This is similar to Equation (11). Therefore, in practical applications, we usually determine $\alpha$ according to the velocity scope of interest. For example, we assume that the velocity scope of interest is $[-v_{max}, v_{max}]$. Then,

we can obtain the maximum value of $K_0$ as $K_{0,max} = \frac{2v_{max}}{\lambda}MT_r$. According to Equation (25), in order to ensure that $K_{0,max}$ can be estimated, $\alpha$ should satisfy the following inequality

$$\begin{cases} \alpha \geq \left\lceil \frac{4v_{max}}{\lambda f_r} \right\rceil \\ \alpha \in \mathbf{N}^* \end{cases}, \tag{27}$$

where $\lceil \cdot \rceil$ is the ceiling operation. In order to have a better estimation accuracy of the velocity, we generally take $\alpha = \left\lceil \frac{4v_{max}}{\lambda f_r} \right\rceil$.

Similar to the analysis of velocity, for the acceleration of the maneuvering target $a_0$, we assume that the acceleration scope of interest is $[-a_{max}, a_{max}]$. Then, we can also obtain the maximum value of $L_{0,max}$, i.e., $L_{0,max} = \frac{a_{max}}{\lambda}M^2T_r^2$. Similarly, based on Equation (25), we can also obtain the constraint of $\beta$ as follows

$$\begin{cases} \left\lceil \frac{2a_{max}M}{\lambda f_r^2} \right\rceil \leq \beta < M \\ \beta \in \mathbf{N}^* \end{cases}. \tag{28}$$

From the analysis of $\beta$ in the previous subsection, it can be known that the SZDCFT would only produce one peak in the $l$ domain under the constraint of $\beta < M$. Here, we default to the acceleration being unambiguous, which is often the case in practice.

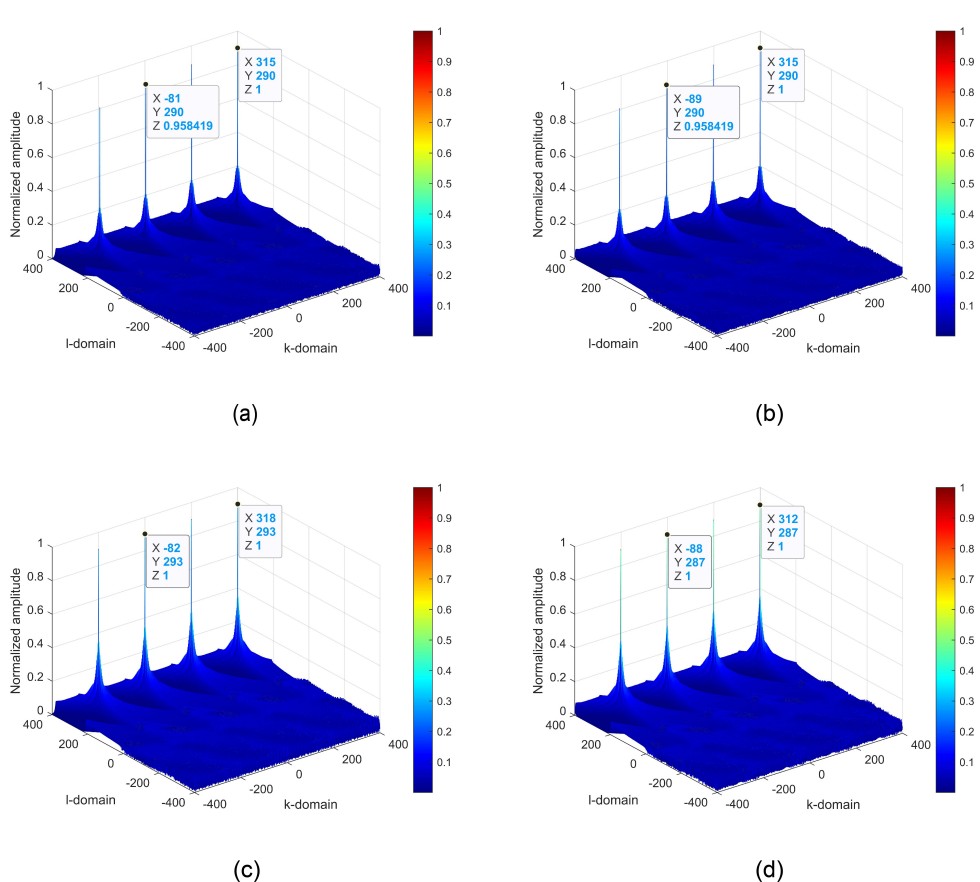

**Figure 3.** Simulation results of Example 3. (**a**) The result of SZDCFT ($\psi_1 = 1.01$). (**b**) The result of SZDCFT ($\psi_2 = 0.99$). (**c**) The result of ZDCFT ($\psi_1 = 1.01$). (**d**) The result of ZDCFT ($\psi_2 = 0.99$).

### 3.4. Product Operation and Coherent Integration

The SZDCFT uncouples the slow time and range frequency, and extends the parameter estimation scope. However, we are unable to identify the true peak when there are multiple peaks in the $k$ domain. To solve this problem, we propose the product SZDCFT method (PSZDCFT).

By re-examining Equations (24) and (25), we can find that: (1) For different range frequency cells, $S_{SZDCFT}(f,k,l)$ has different implied periods in the $k$ domain ($M_{kf} = \frac{M}{\alpha\psi(f)}$), which means that when there are multiple peaks, the interval between peaks in the two-dimensional spectrum varies with the change in the range frequency. (2) The true peak position is always constant in the two-dimensional spectrum, i.e., $(K_0/\alpha, L_0/\beta)$, while the positions of the false peaks are changed with the range frequency. The comparison between Figure 3a,b supports the above assertions very well.

Therefore, it naturally occurs to us to perform a product operation on the SZDCFT results for all range frequency cells, which can curb the false peaks and heighten the true peak. Then, the PSZDCFT method is defined as

$$
\begin{aligned}
S_{PSZDCFT}(k,l) &= \text{Product}[S_{SZDCFT(f,k,l)}] \\
&= \prod_{f=-B/2}^{B/2} S_{SZDCFT}(f,k,l) \\
&\approx \sigma_{PSZDCFT} \cdot \delta(k - K_0/\alpha), \quad l = L_0/\beta,
\end{aligned}
\tag{29}
$$

where $\sigma_{PSZDCFT}$ is the amplitude of accumulation.

From Equation (29), it can be seen that there will be only true peak in the two-dimensional spectrum, while the false peaks are eliminated. Then, based on the peak location, we can obtain the velocity and acceleration of the maneuvering target as follows

$$
\begin{cases}
(k_{max}, l_{max}) = \arg\max_{(k,l)} |\text{Product}\{SZDCFT[S_c(f,m)]\}| \\
\hat{v}_0 = -\frac{\alpha\lambda f_r k_{max}}{2M} \\
\hat{a}_0 = -\frac{\beta\lambda f_r^2 l_{max}}{M^2}
\end{cases}
\tag{30}
$$

After obtaining the estimated velocity and acceleration of the maneuvering target, we can construct the phase compensation function to eliminate the RCM and DFCM, i.e.,

$$
H_{PCF}(f, \eta_m) = \exp\left(j\frac{4\pi}{c} f\hat{v}_0\eta_m\right) \exp\left(j\frac{2\pi}{c}(f + f_c)\hat{a}_0\eta_m^2\right).
\tag{31}
$$

Finally, we can achieve the coherent integration by applying the IFFT and FFT with respect to $f$ and $\eta_m$.

$$
\begin{aligned}
S_{CI}(\hat{t}, f_{\eta_m}) &= \text{FFT}_{\eta_m}\left\{\text{IFFT}_f[S_c(f, \eta_m) \cdot H_{PCF}(f, \eta_m)]\right\} \\
&= \sigma_{CI} \text{sinc}\left[B\left(\hat{t} - \frac{2R_0}{c}\right)\right] \\
&\times \text{sinc}\left[CPI\left(f_{\eta_m} + \frac{2v_{res}}{\lambda}\right)\right],
\end{aligned}
\tag{32}
$$

where $\sigma_{CI}$ and $CPI$ denote the amplitude and coherent processing interval, respectively. $f_{\eta_m}$ is the folded Doppler frequency with respect to $\eta_m$.

## 4. Analysis of the Proposed Method

### 4.1. PSZDCFT for Multi-Targets

The SZDCFT is a linear transform, which can avert the interference of cross terms. Therefore, for multi-target scenarios, the proposed approach is still applicable. It should be noted that the product operation may annihilate the weak target when there are significant differences in the echo amplitudes between targets. Fortunately, the "CLEAN" technique [40] can be used, to remove the strong target effect.

### 4.2. Implementation of SZDCFT

The SZDCFT of $S_c(f, m)$ can be rewritten as follows

$$
\begin{aligned}
S_{SZDCFT}(f, k, l) &= \frac{1}{\sqrt{M}} \sum_{m=-M/2}^{M/2-1} S_c(f, m) W_M^{\psi(f)\left(\alpha km + \beta \frac{l}{M^2} m^2\right)} \\
&= \frac{1}{\sqrt{M}} \sum_{m=-M/2}^{M/2-1} \left[ S_c(f, m) W_M^{\psi(f)\beta \frac{l}{M^2} m^2} \right] \\
&\quad \times \exp\left( -j\frac{2\pi}{M} \psi(f)\alpha km \right).
\end{aligned}
\tag{33}
$$

Equation (33) shows that, for each fixed $l$, $S_{SZDCFT}(f, k, l)$ is the scaled Fourier transform (SFT) of the signal $\left[ S_c(f, m) W_M^{\psi(f)\beta \frac{l}{M^2} m^2} \right]$. As we know, the SFT can be efficiently implemented by FFT-based chirp-z transform (CZT).

Assuming a discrete signal $x(n)$, $n = -N/2, \ldots, N/2 - 1$, the SFT of $x(n)$ is defined as follows

$$
\begin{aligned}
X(\xi k) &= \sum_{n=-N/2}^{N/2-1} x(n) \exp\left( -j\frac{2\pi}{N} \xi nk \right) \\
&= \sum_{n=-N/2}^{N/2-1} x(n) \tilde{W}^{nk},
\end{aligned}
\tag{34}
$$

where $k = -N/2, \ldots N/2 - 1$ and $\tilde{W} = \exp\left(-j\frac{2\pi}{N}\xi\right)$. $\xi$ is the scale factor. Then, the Bluestein equation is applied here, i.e.,

$$
nk = \frac{1}{2}\left[ n^2 + k^2 - (n-k)^2 \right].
\tag{35}
$$

Inserting Equation (35) into Equation (34), one obtains

$$
\begin{aligned}
X(\xi k) &= \sum_{n=-N/2}^{N/2-1} x(n) \tilde{W}^{nk} \\
&= \tilde{W}^{\frac{k^2}{2}} \sum_{n=-N/2}^{N/2-1} \left[ x(n) \tilde{W}^{\frac{n^2}{2}} \right] \tilde{W}^{\frac{-(k-n)^2}{2}} \\
&= \tilde{W}^{\frac{k^2}{2}} \left\{ \left[ x(n) \tilde{W}^{\frac{n^2}{2}} \right] \circledast \tilde{W}^{-\frac{n^2}{2}} \right\},
\end{aligned}
\tag{36}
$$

where $\circledast$ represents convolution. Figure 4 illustrates the calculation process of Equation (36). Thus, the SFT of $N$ points can be efficiently realized by complex multiplication, FFT and IFFT, which requires a computational load of $O(3N\log_2 N)$.

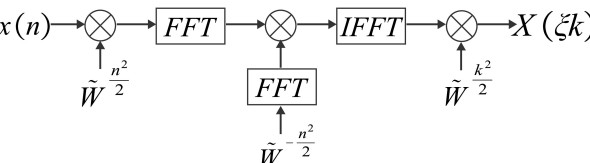

**Figure 4.** The calculation process of CZT using FFT.

### 4.3. Computational Complexity

The computational complexities of GRFT, RLVD, ACCF, PC-TRT, FAF-LVD, and the proposed method, are analyzed and compared in this subsection. Suppose $M$, $N$, $N_v$, and $N_a$ represent the number of pulses, range cells, searching velocity, and searching acceleration, respectively.

The GRFT accomplishes coherent integration through a three-dimensional parameter search. Thus, its computational load is about $O(NN_vN_aM)$.

The RLVD firstly obtains the target trajectory through a three-dimensional search, which requires a computational cost of $O(NN_vN_a)$, and then accomplishes coherent integration via LVD, with a computational cost of $O(3M^2log_2M)$. Therefore, its total computational complexity is $O(3NN_vN_aM^2log_2M)$.

For the ACCF method, it uses an adjacent cross-correlation operation to correct the RCM and eliminate the DFCM, which requires a computational cost of $O(2MNlog_2N)$. Then, the computational cost required to achieve coherent integration is $O(MN(log_2M + log_2N))$. Hence, its total computational cost is about $O(MN(3log_2N + log_2M))$.

As for PC-TRT, it decouples the slow time and range frequency by constructing the phase compensation function with the searched velocity, and performing the time reversal transform, and finally realizes coherent integration using IFFT and FFT. Therefore, its computational cost is about $O(N_vMNlog_2(MN))$.

The main calculations of FAF-LVD contain second-order KT $O(4MNlog_2M)$, FAF $O(MNlog_2N)$, CZT-based SFT $O(3MNlog_2M)$, phase compensation $O(MNlog_2N)$, and LVD $O(3M^2log_2M)$. Therefore, its total computational burden is about $O(3M^2log_2M + MN(7log_2M + 2log_2N))$.

For each fixed range cell of the proposed method, the computational costs required by the SZDCFT operation and the product operation are $O(M^2 + 3M^2log_2M)$ and $O(M^2)$, respectively. Therefore, the overall computational load of the PSZDCFT, is in the order of $O(NM^2(2 + 3log_2M))$.

Table 1 shows the computational complexities of the above methods. We assume that $N = N_v = N_a = M$. Then, Figure 5 depicts the computational complexity curves. The RLVD and GRFT take up too much time and are not conducive to real-time processing. It can be seen that although the computational complexity of PC-TRT remains of the same order of magnitude as that of the present method, i.e., $O(M^3log_2M)$, the PC-TRT has an inferior detection performance under the condition of low SNR, which is due to the fact that the TRT operation loses the bulk of the signal energy. The FAF-LVD and ACCF obtain the lowest computational loads via the correlation operation, but there are also noticeable drops in the anti-noise performance. Hence, it can be inferred that the proposed method accomplishes a better compromise between computational burden and anti-noise performance.

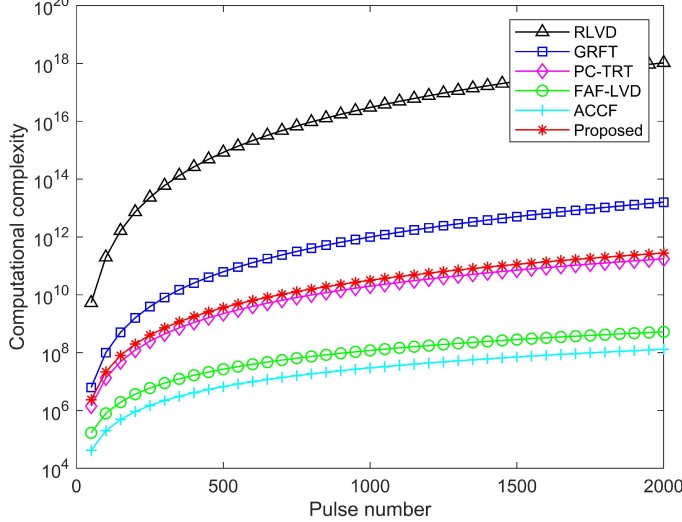

**Figure 5.** Computational complexity curves.

**Table 1.** Computational complexity comparison.

| Methods | Computational Complexity |
|---|---|
| GRFT | $O(N_v N_a M N)$ |
| RLVD | $O(3N N_v N_a M^2 log_2 M)$ |
| ACCF | $O(MN(2log_2 N + log_2 M))$ |
| PC-TRT | $O(N_v M N log_2(MN))$ |
| FAF-LVD | $O(3M^2 log_2 M + MN(7log_2 M + 2log_2 N))$ |
| Proposed | $O(NM^2(2 + 3log_2 M))$ |

*4.4. Procedure of the Proposed Method*

The flow chart of the proposed method is shown in Figure 6 and its main procedures are described in the following subsection.

Step 1: Perform PC on the received baseband signal $s_r(\hat{t}, \eta_m)$, then employ the FFT along the range dimension, to obtain $S_c(f, \eta_m)$.

Step 2: For a fixed range frequency $f$, apply the SZDCFT on $S_c(f, \eta_m)$, to obtain $S_{SZDCFT}(f, k, l)$.

Step 3: Determine the scope of the next range frequency, $f$. If $f \in [-B/2, B/2]$, execute step 2. If $f \notin [-B/2, B/2]$, execute product operation, to obtain $S_{PSZDCFT}(k, l)$.

Step 4: Estimate the target's velocity and acceleration according to Equation (30), and then construct the phase compensation function based on Equation (31).

Step 5: Perform the IFFT and FFT along the range dimension and slow-time dimension, respectively, to accomplish coherent integration.

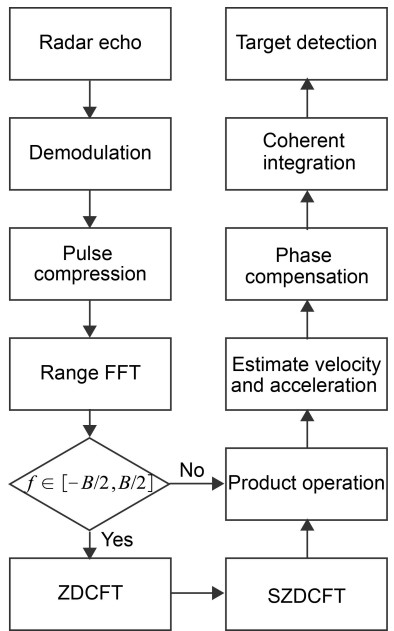

**Figure 6.** The flowchart of the proposed method.

## 5. Numerical Results

In this section, the validity and reliability of the presented method are verified by numerical simulations, with the use of the Matlab numerical software; the radar system parameters are listed in Table 2.

*5.1. Coherent Integration for a Single Target*

Figure 7 illustrates the coherent integration ability for a single target , where the motion parameters of the target are: initial range cell number $n_0 = 451$st, radial velocity $v_0 = 120$ m/s, and radial acceleration $a_0 = 45$ m/s$^2$. The input SNR after PC is 20 dB. Based on Equations (27) and (28), we take $\alpha = 4$ and $\beta = 2$, to extend the estimation

scopes of velocity and acceleration to $[-150, 150)$ m/s and $[-75, 75)$ m/s$^2$, respectively. Figure 7a displays the target motion trajectory after PC. It is apparent that serious RCM occurs to the target, due to the high velocity and acceleration. Figure 7b provides the result of MDCFT when $f = -4$ MHz. It can be seen that the MDCFT fails because the motion parameters exceed the estimation scopes of MDCFT. Figure 7c,d show the results of SZDCFT when $f = \pm 4$ MHz. It can be observed that there are four peaks in each two-dimensional spectrum diagram. The true peak positions are the same, i.e., $(\hat{v}_0, \hat{a}_0) = (120, 45)$, while the false peak positions are interlaced. The result of PSZDCFT is depicted in Figure 7e. As predicted, the false peaks are removed, while the true peak is reinforced and recognized. Then, based on the location of the true peak, we obtain the estimated velocity $\hat{v}_0$ and acceleration $\hat{a}_0$ as being 120 m/s and 45 m/s$^2$, respectively. Finally, the estimated parameters are used to construct a phase compensation function, to eliminate the RCM and DFCM, and then the target energy can be well focused, in Figure 7f.

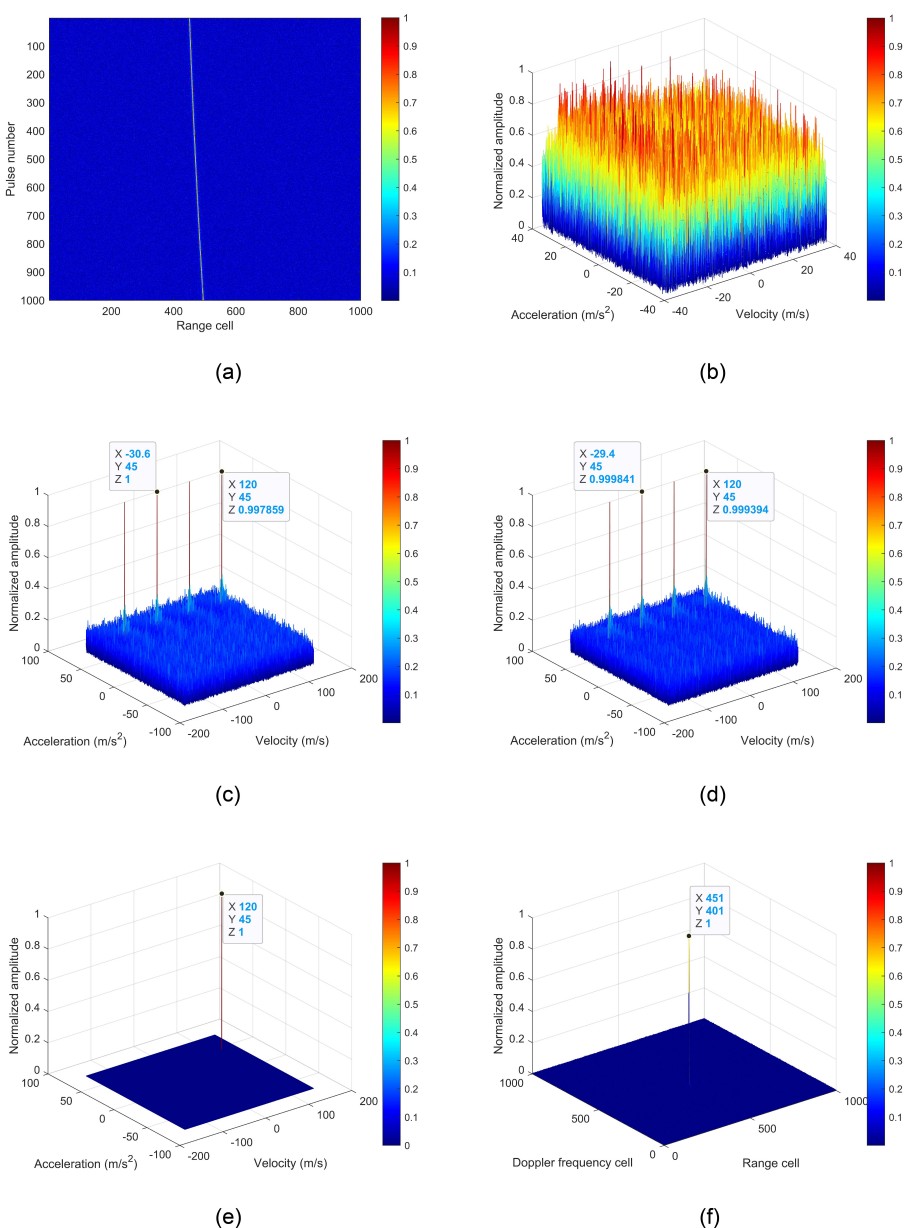

**Figure 7.** Coherent integration for a single target at SNR = 20 dB. (**a**) The result of PC. (**b**) The result of MDCFT at $f = -4$ MHz. (**c**) The result of SZDCFT at $f = -4$ MHz. (**d**) The result of SZDCFT at $f = 4$ MHz. (**e**) The result of PSZDCFT. (**f**) The result of coherent integration.

**Table 2.** Simulation parameters of radar.

| | |
|---|---|
| Carrier frequency, $f_c$ | 1 GHz |
| Bandwidth, $B$ | 10 MHz |
| Sample frequency, $f_s$ | 20 MHz |
| Pulse repetition frequency, *PRF* | 500 Hz |
| Pulse duration, $T_p$ | 20 µs |
| Number of pulses, $M$ | 1000 |

Figure 8 intuitively demonstrates the excellent ability of the product operation with respect to focusing energy and estimating parameters, in a low SNR environment. The input SNR is $-8$ dB after PC. Other parameters are identical to those in Figure 7. Figure 8a shows the result after PC, from which it can be observed that the target trajectory is completely annihilated in the noise. Figure 8b provides the result of SZDCFT at $f = 0$ MHz. It can be noticed that the peak is not visible, which is not favorable for parameter estimation. In this regard, we perform a product operation for the SZDCFT results along the range frequency cells, to accumulate the signal energy dispersed in different range frequency cells, which is illustrated in Figure 8c. Comparing Figure 7c,d and Figure 8b, it is easy to see that the product operation still has a remarkable parameter estimation ability in the low SNR environment, which is also confirmed in Section 5.4. On this basis, coherent integration can naturally also be accomplished well, in Figure 8d.

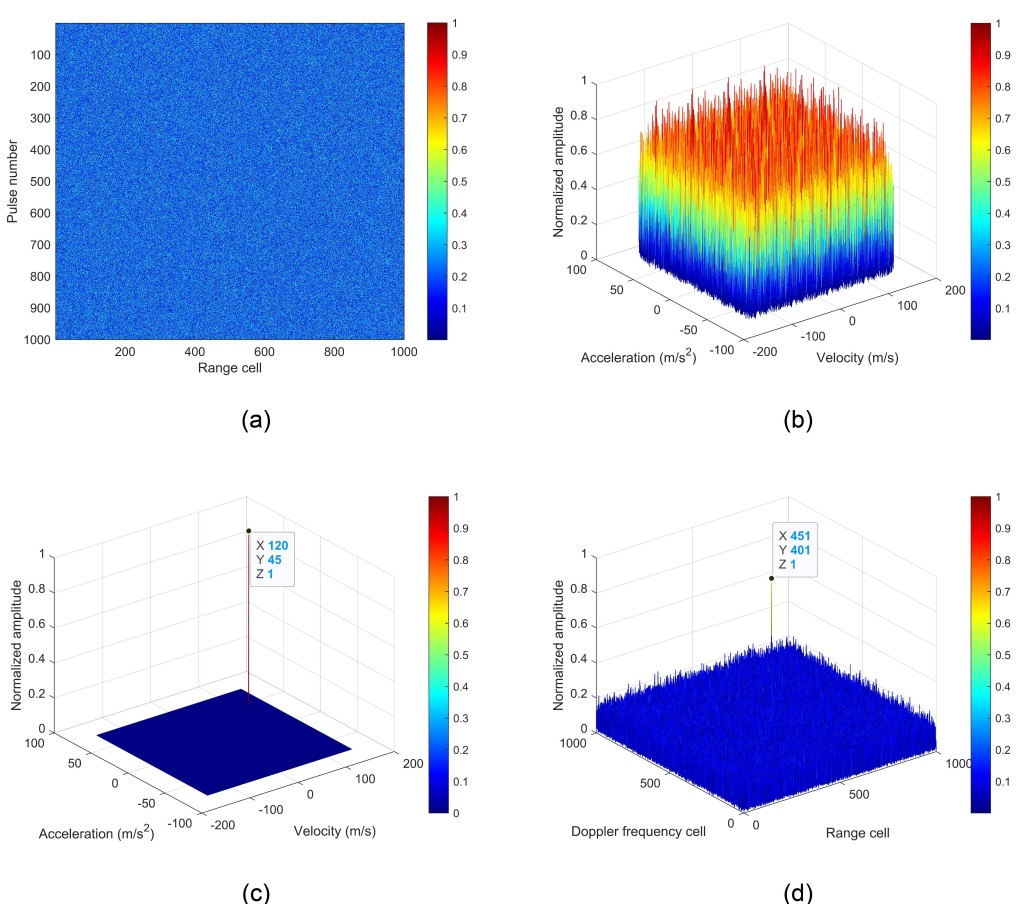

**Figure 8.** The role of product operation in a low SNR environment (**a**) The result of PC. (**b**) The result of SZDCFT at $f = 0$ MHz. (**c**) The result of PSZDCFT. (**d**) The result of coherent integration.

### 5.2. Coherent Integration for Multi-Targets

We also evaluate the coherent integration capability of the presented approach for multiple targets in Figure 9. The motion parameters of the three maneuvering targets are given in Table 3. Again, after PC, the input SNR is 20 dB. Similarly, depending on the motion parameters of the three maneuvering targets, we choose $\alpha = 4$ and $\beta = 2$. Three curved trajectories can be observed in Figure 9a, which indicates that the three targets undergo severe RCMs. Figure 9b,c depict the results of SZDCFT when $f = \pm 4$ MHz. We can find that after performing the SZDCFT operation, the true peak positions are independent of the range frequency, while the false peak positions are coupled with the range frequency. Then, we perform a product operation on the results of SZDCFT for all range frequency cells, to obtain three spikes, shown in Figure 9d. Based on the three peak locations, we obtain the motion parameters of targets A, B, and C, i.e., $\hat{v}_A = 120$ m/s, $\hat{a}_A = 45$ m/s$^2$, $\hat{v}_B = -108$ m/s, $\hat{a}_B = -30$ m/s$^2$, $\hat{v}_C = 117$ m/s, and $\hat{a}_C = 42$ m/s$^2$. Finally, the coherent integration results for targets A, B, and C are provided in Figure 9e,f,g, respectively.

**Table 3.** Motion parameters of three maneuvering targets.

| Motion Parameters | Target A | Target B | Target C |
|---|---|---|---|
| Initial range cell number | 451st | 476st | 501st |
| Radial velocity | 120 m/s | −108 m/s | 117 m/s |
| Radial acceleration | 45 m/s$^2$ | −30 m/s$^2$ | 42 m/s$^2$ |
| SNR (after PC) | 20 dB | 20 dB | 20 dB |

### 5.3. Detection Performance

In this section, we evaluate the target detection capability of the presented method through Monte Carlo experiments. As a comparison, five other representative methods, i.e., RLVD, GRFT, FAF-LVD, PC-TRT, and ACCF, are also simulated. The constant false alarm rate (CFAR) technique, with false alarm rate $P_{fa} = 10^{-6}$, is used. The input SNR after PC varies from $-25$ to 10 dB, with a step of 1 dB. For each SNR value, 500 independent simulations are run. Figure 10 depicts the detection probability curves.

It can be seen that the GRFT has the best detection performance, thanks to the multidimensional parameter search. The RLVD achieves a detection performance close to GRFT's, again attributed to the parameter search and excellent LVD operation. Using GRFT as a baseline, the proposed method suffers an SNR loss of about 3 dB, when the detection probability $p_d$ = 0.9. This is because the proposed method estimates the parameters by the product operation, which introduces more random errors compared with GRFT, which is based on the maximum likelihood estimation (MLE), in an extremely low SNR environment. For a clearer explanation, Figure 11 offers the results of parameter estimation for the two methods at an extremely low SNR, of $-12$ dB. The simulation parameters are the same as those in Figure 7. Figure 11a gives the results of the proposed method by the product operation. It can be seen that at extremely low SNRs, multiple randomly distributed peaks appear in the plot after the product operation, due to the influence of noise, which deteriorates the parameter estimation performance. As a contrast, the results of GRFT, based on the MLE, is displayed in Figure 11b. Fortunately, due to the fact that the proposed approach is a linear transform with no energy loss, it can be observed from Figure 10 that the method has a superior anti-noise performance at low SNRs, compared with FAF-LVD, PC-TRT, and ACCF. Therefore, considering the processing time and detection performance, we can deduce that the presented approach is more applicable for maneuvering targets in low SNR environments.

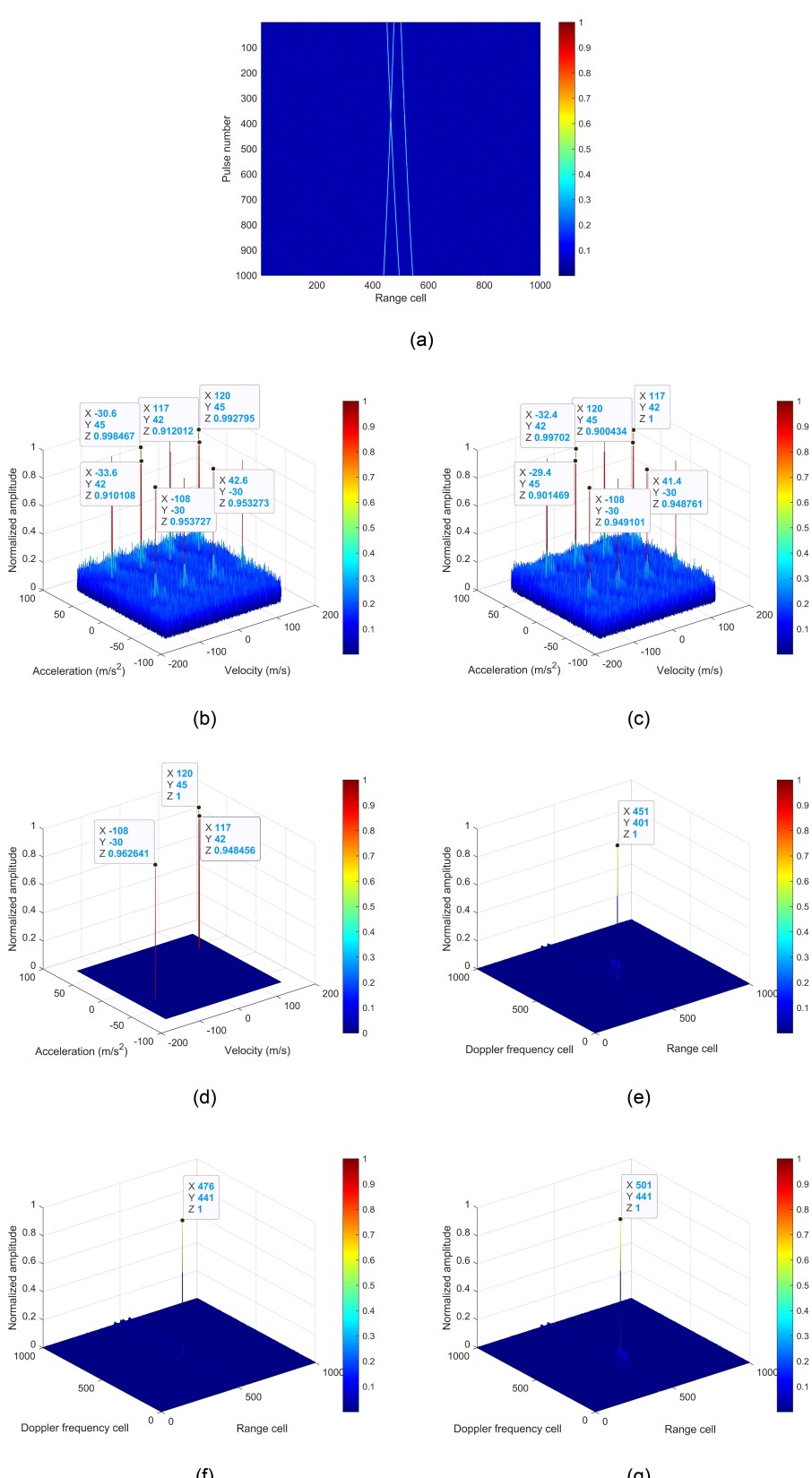

**Figure 9.** Coherent integration for multi-targets. (**a**) The result of PC. (**b**) The result of SZDCFT at $f = -4$ MHz. (**c**) The result of SZDCFT at $f = 4$ MHz. (**d**) The result of PSZDCFT. (**e**) The result of coherent integration for target A. (**f**) The result of coherent integration for target B. (**g**) The result of coherent integration for target C.

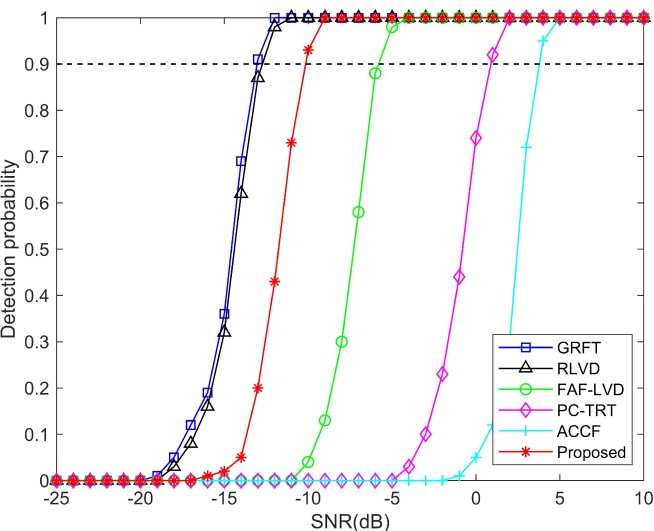

**Figure 10.** Detection probability curves of different methods.

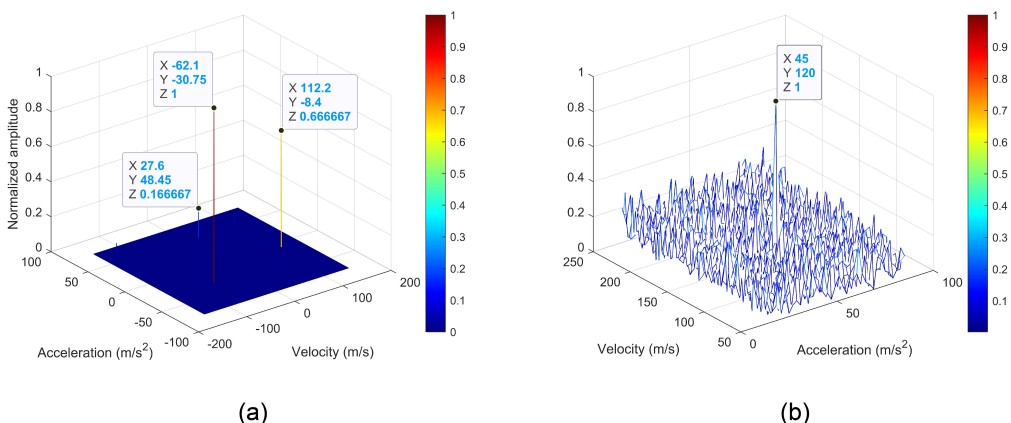

**Figure 11.** The results of parameter estimation for the proposed method and GRFT at SNR = −12 dB. (**a**) The result of of parameter estimation for the proposed method. (**b**) The result of of parameter estimation for GRFT.

### 5.4. Parameter Estimation Performance

In this section, we evaluate the parameter estimation performance of GRFT, FAF-LVD, PC-TRT, and the presented method, by Monte Carlo experiments. For the sake of comparison, the ranges of the estimated motion parameters by the four approaches are set to be identical. The input SNR after PC varies from −25 to 10 dB, in steps of 1 dB. For each SNR value, 500 independent experiments are carried out. The root mean square error (RMSE) is used as the evaluation criterion. The RMSE curves for velocity and acceleration are plotted in Figure 12.

The GRFT is demonstrated to have optimal parameter estimation performance, owing to the MLE. The presented method has a performance loss of about 3 dB compared to GRFT, due to the product operation. The PC-TRT has the worst RMSE, because it only uses one time slice for nonlinear correlation, which loses much signal energy. Although the FAF-LVD can reduce the energy loss through a variable latency, its RMSE is dramatically increased when the SNR is less than −5 dB, compared to the presented method. Thus, the presented approach acquires a better equilibrium between computational cost, detection performance, and parameter estimation.

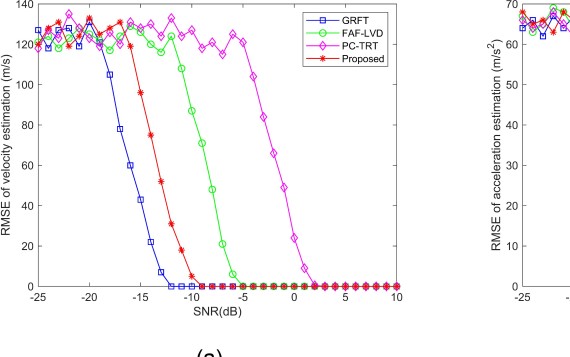
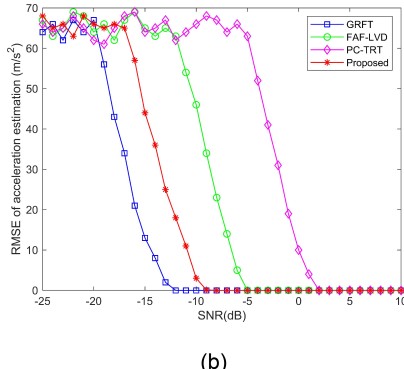

(a) (b)

**Figure 12.** Parameter estimation performance of different methods. (**a**) RMSE of velocity estimation. (**b**) RMSE of acceleration estimation.

## 6. Conclusions

In this paper, a new coherent integration method, based on PSZDCFT, is proposed for maneuvering targets, which effectively addresses the RCM and DFCM. For maneuvering targets with velocity ambiguity and excessive acceleration, the proposed method introduces a zoom operation into MDCFT, to extend its parameter estimation ranges, and it is interesting to note that this operation is also suitable for uniform motion targets with velocity ambiguity. Considering the coupling of range frequency and slow time, the scale operation is combined with ZDCFT to decouple, which not only obtains the correct parameter estimates, but also paves the way for the product operation, excluding false peaks. Naturally, the product operation is employed along the range frequency, which can inhibit false peaks and accumulate energy dispersed in different range cells. Finally, by constructing the phase compensation function with the estimated parameters, coherent integration is achieved. The computational complexity analysis shows that the computational cost of the proposed method is comparatively small, thanks to its fast implementation using the CZT-based SFT. Intensive numerical simulation results demonstrate that the proposed method is robust in a low SNR environment, due to the fact that the proposed method is a linear transform with no energy loss and no cross terms for multiple targets. Therefore, the proposed method obtains a superior equilibrium between computational complexity and detection performance, and is more suitable for maneuvering target detection in comparison with most current methods.

**Author Contributions:** Conceptualization, L.X. and H.G.; methodology, L.X. and H.G.; software, L.X. and T.L.; validation, L.X., T.L. and B.F.; formal analysis, L.X. and B.F.; resources, L.L.; writing—original draft preparation, L.X.; writing—review and editing, L.X. and H.G.; supervision, B.F. and L.L.; project administration, L.X., L.L. and T.L.; funding acquisition, H.G. and L.L. All authors have read and agreed to the published version of the manuscript.

**Funding:** This work was supported in part by the National Natural Science Foundation of China (under Grant 61671333), in part by the Natural Science Foundation of Hubei Province (under Grant 2014CFA093), in part by the Fundamental Research Funds for the Central Universities (under Grant 2042019K50264, Grant 2042019gf0013, and Grant 2042020gf0003), and in part by the Fundamental Research Funds for the Wuhan Maritime Communication Research Institute (under Grant 2017J-13 and Grant KCJJ2019-05).

**Institutional Review Board Statement:** Not applicable.

**Informed Consent Statement:** Not applicable.

**Data Availability Statement:** The data presented in this study are available on request from the corresponding author. The data are not publicly available due to the fact that it is currently privileged information.

**Conflicts of Interest:** The authors declare no conflict of interest.

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
