# Peer review of "Radar Maneuvering Target Detection Based on Product Scale Zoom Discrete Chirp Fourier Transform"

_remotesensing, doi:10.3390/rs15071792_

Round 1

Reviewer 1 Report

The authors proposed a coherent integration method to detect high maneuvering targets in low SNR environments. This paper showed interesting research and introduced the method systematically. However, I have some comments and questions:

1. In introduction, the contribution of the method contains good performance for multi-target scenarios and extending the estimation scopes of velocity and acceleration, but they were not fully considered in the background. Please add explanation for whether the existing studies in the background are suitable for multi-target scenarios and estimating the excess motion parameters.

2. In Section 3.2, the Zoom DCFT method is well described. But how to choose the parameter alpha and beta? Do they need any prior information of the centroid frequency and chirp rate? Please supplement the explanation.

3. Figure 1 and Figure 2 shows the simulation results of Example 1 and Example 2, but the results could be better explained. Two peaks are marked in the Figure 1b and Figure 2a, but only the true peak's locations are explained in the text. Please add the theoretical values of the marked false peak.

4. Figure 6 shows a clear flowchart of the proposed method, but it is oversize. please check how to make the size smaller so that the text in the figure is no larger than in context.

Author Response

Dear Reviewer1,

We have tried our best to revise our manuscript according to your suggestions, and the revised parts are marked in yellow. The point-to-point responses are submitted as a PDF.

Yours Sincerely,

Lang Xia

Reviewer 2 Report

I have read the paper on a novel approach for better balancing computational cost and anti-noise performance based on PSZDCFT, and I believe it has great potential. However, I would like to offer a few suggestions to enhance the paper's quality:

1. Provide more details on how the analyses were performed, including which software was used. It would also be helpful to provide the algorithm as supplementary material or on a developer platform.

2. Avoid using expressions that assume common knowledge, such as "As we all know," "well-known," "obvious," etc.

3. Revise the highlights at the end of the introduction section (lines 93 to 104), as they appear to be the conclusion or main results of the paper.

4. Review the examples mentioned in the method section, as they appear to be results. If they are not, consider providing these examples as supplementary material.

5. Clearly distinct what is a method and how the results were evaluated.

6. Consider including a discussion section that highlights the points made in the introduction, demonstrating that the method is more suitable for this type of analysis. This section should also clarify how the method performed better in comparison to others.

7. The conclusions are unclear. It would be more beneficial to state explicitly what results were found and the conclusions drawn from them.  

Overall, I believe that implementing these suggestions will help to enhance the quality of the paper and make it more impactful in the field.

Author Response

Dear Reviewer2,

We have tried our best to revise our manuscript according to your suggestions, and the revised parts are marked in yellow. The point-to-point responses are submitted as a PDF.

Yours Sincerely,

Lang Xia

Reviewer 3 Report

This paper proposes a novel maneuvering targets detection method to solve the problem that is generated by the range cell migration (RCM) and Doppler frequency cell migration (DFCM) during long time coherent integration. Firstly, the traditional Chirp Fourier transform is modified and ameliorated to estimate the motion parameters of the maneuvering targets. Then the phase compensation based on estimated parameters is utilized to achieve the target detection. Through the illustration of simulation experiments, the proposed method can detect the maneuvering target in the multiple target scenario. There is a certain amount of innovation in this paper. However, there are also some details need to be improved:

1.     The statement PC (Pulse Compression) in Line 123 is not accurate. Usually, the PC can be realized by matched filtering that including two times FFT and once IFFT processing. However, the equation (4) is still in the range frequency and slow time domain, thus the PC processing is not completed.

2.     From the flow chart of the proposed method in Figure 6, it can be seen that the motion parameters estimation is before the target detection. In general, we all know that the target is detected firstly and then some motion parameters can be estimated from radar echo. Therefore, there is a question whether it is logical?

3.     The Figure 8 gives the result of low SNR experiments, which is similar to Figure 7. To offer more intuitionistic and brief illustration for readers, the subsection 5.1 can select a lower SNR environment, for example 3 dB, and avoid the two homogenous experiments. Moreover, the subsection 5.3 demonstrates the detection performance of the proposed method under different SNR.

Author Response

Dear Reviewer3,

We have tried our best to revise our manuscript according to your suggestions, and the revised parts are marked in yellow. The point-to-point responses are submitted as a PDF.

Yours Sincerely,

Lang Xia

Reviewer 4 Report

This paper proposes a new PSZDCFT method for maneuvering target detection. The method works well on the cases shown and I believe the innovation of this manuscript is enough. I have some questiones:

1. Will adding two scale parameters alpha and beta only increase the number of target peaks? Will it lead to the decline of estimation accuracy?

2. Why doesn't the computational complexity compare with the MDCFT method?

3. Why is the maximum speed range in Figure 7b only 40 m/s? Is the detection range of MDCFT limited to this value?

4. The speed direction of the two targets in Table 3 is opposite. If the direction and the speed value of the two targets are relatively close, what will the final performance be?

5. The experimental part in 5.3 and 5.4 is not compared with MDCFT. Is it because the results of the proposed method and MDCFT are the same?

Author Response

Dear Reviewer4,

We have tried our best to revise our manuscript according to your suggestions, and the revised parts are marked in yellow. The point-to-point responses are submitted as a PDF.

Yours Sincerely,

Lang Xia

Round 2

Reviewer 2 Report

The authors provided sufficient and satisfactory responses to all the questions posed.